# OCT and ERG Techniques in High-Throughput Phenotyping of Mouse Vision

**DOI:** 10.3390/genes14020294

**Published:** 2023-01-22

**Authors:** Jiri Lindovsky, Marcela Palkova, Viktoriia Symkina, Miles Joseph Raishbrook, Jan Prochazka, Radislav Sedlacek

**Affiliations:** Czech Centre for Phenogenomics, Institute of Molecular Genetics, Czech Academy of Sciences, 252 50 Vestec, Czech Republic

**Keywords:** phenotyping, vision, OCT, ERG, IMPC

## Abstract

The purpose of the study is to demonstrate coherent optical tomography and electroretinography techniques adopted from the human clinical practice to assess the morphology and function of the mouse retina in a high-throughput phenotyping environment. We present the normal range of wild-type C57Bl/6NCrl retinal parameters in six age groups between 10 and 100 weeks as well as examples of mild and severe pathologies resulting from knocking out a single protein-coding gene. We also show example data obtained by more detailed analysis or additional methods useful in eye research, for example, the angiography of a superficial and deep vascular complex. We discuss the feasibility of these techniques in conditions demanding a high-throughput approach such as the systemic phenotyping carried out by the International Mouse Phenotyping Consortium.

## 1. Introduction

The laboratory mouse is a well-established animal model important to vision research. This small rodent is easy to breed and maintain in the laboratory, reproduces rapidly, and reaches adulthood in twelve weeks, while one-year-old individuals can already be considered as adequate models for age-related diseases. Moreover, the complete mouse genome has been sequenced and a comparison to that of humans showed that 99% of homologous protein-coding genes exist in different copy numbers in the two species [1]. Together with modern gene-editing techniques, this laboratory animal represents an excellent model organism for large-scale studies of the gene–function relation, the causative role of genes in disease, or gene function during aging. Such an activity is currently carried out by the International Mouse Phenotyping Consortium (IMPC, www.mousephenotype.org), which is an international effort by 21 research institutions to identify the function of every protein-coding gene in the mouse genome. To achieve this, the IMPC is systematically knocking out each of the roughly 20,000 genes that make up the mouse genome. Subsequently, the mice undergo standardized physiological tests (phenotyping) across a range of biological systems in order to infer gene function before the data is made freely available to the research community. Phenotyping tests are organized into pipelines that cover several periods of the mouse life (www.mousephenotype.org/impress/PipelineInfo?id=44 (accessed on 1 December 2022)).

Spectral domain optical coherent tomography (OCT) can noninvasively image retinal structures in vivo due to the distinct light reflectance of each retinal layer. OCT is fast and provides 3D scans with micrometer resolution [2,3,4,5,6]. Lipofuscin deposits in the retinal pigmented epithelium, e.g., caused by metabolic changes, can be detected due to their autofluorescence by a confocal scanning laser module using a laser in the blue wavelength range. Moreover, the detection of moving particles may be used for the imaging of perfused retinal vascular plexuses and the choroid and reveal even small changes in the microvasculature. Being noninvasive, these imaging techniques enable repeated examinations in longitudinal studies to monitor the onset and progression of diseases or dynamic changes in the retinal structure during aging.

Electroretinography, ERG, for review see [7,8], measures the electrical responses of the retina evoked by light stimulation from the eye surface. It is a summed activity of multiple retinal cell populations and consists of overlapping positive and negative potentials that originate from different stages of retinal processing. A signal is obtained under various light conditions, stimulation intensities, and timing protocols to individually assess the function of rods, cones, ON bipolar cells, the pigment epithelium, and others. The first major negative peak visible in the ERG response to a flash stimulation is the wave a. It reflects a change in the electric potential of photoreceptors resulting from the detection of light [9]. The second large wave is the positive-going wave b, which corresponds to the activation of the ON bipolar cells [10]. ERG is widely used in medicine; the method is noninvasive and can be utilized in animals in a similar way as in humans [11,12].

Commercially available devices for optical imaging and the electrophysiology of vision are often developed for clinical use in humans. In principle, they can be very well applied in animal research as well; however, the experimenter should be aware of the specific characteristics of the used model. The mouse eye is much smaller (<4 mm, [13]) than the human eye, has a ball-shaped rather than biconvex lens [14], and a thinner retina (~230 µm, [15]). Photoreceptor composition is also different, displaying dichromacy with spectral sensitivity shifted towards the ultraviolet range [16,17], a smaller proportion of cones [18], and the absence of a macula [19]. Additionally, the mouse retina is most commonly examined under general anesthesia. Here, we demonstrate that, in spite of the above-mentioned differences, commercial OCT and ERG devices may be applied in mouse research in high-throughput screening conditions.

## 2. Materials and Methods

### 2.1. Animals

Experiments were carried out in wild-type and mutant lines of both sexes of C57Bl/6NCrl mice born inhouse (in the Czech Centre for Phenogenomics). Mice were examined at the ages of 10, 15, 30, 50, 70, and 100 weeks. Animals were housed in individually ventilated cages (Tecniplast, Buguggiate, Italy) under a standard light condition (LD 12:12) at temperature of 22.5 °C and 55% humidity with food and water *ad libitum*. All experiments were carried out under general anesthesia (i.m., Tiletamine + Zolazepam, 30 + 30 mg/kg, supplemented with 3.2 mg/kg Xylazine). Anaesthetized mice were kept on a heating pad throughout the experiment, their pupils were dilated with a drop of 0.5% Atropine (Ursapharm, Prague, Czech Republic), and the eyes were protected against drying using a small amount of transparent eye gel (Vidisic, Bausch&Lomb, Prague, Czech Republic). Experiments were approved by the Animal Care and Use Committee of the Czech Academy of Sciences.

### 2.2. OCT

For the imaging of the posterior eye segment/retina, we used a spectral domain–optical coherent tomography with a standard 30° lens (SD-OCT—Heidelberg-Engineering, Heidelberg, Germany) equipped for animal experiments with an additional +25 dpt lens (Figure 1A). The mice were fitted with +100 dpt contact lenses (Roland Consult, Brandenburg, Germany) and positioned with the scanned eye aiming at the OCT camera so that the head of the optic nerve (OD) appeared in the center of the image. A volume scan comprising of at least 55 B-scans (cross-sections) was collected for both eyes with at least 32 images per B-scan. This setting corresponded to a distance of 120 µm between cross-sections. The automatic segmentation algorithm of the Heidelberg OCT software was used to create a color-coded map of the retinal thickness and the total retinal thickness (TRT) was calculated by the software for all quadrants in the inner and outer ring areas (Figure 2A). The software did not allow to save any numeric data; in order to quantify the retinal thickness automatically, we used screenshots and retrieved numbers using Google Cloud and the text recognition procedure from Cloud Vision API. The automatic segmentation was corrected in obvious erroneous points, and the TRT was calculated again. To analyze individual retinal layers, we set boundaries of the retina (Figure 2B, most right) and its main layers in fully manual mode and assessed the thickness manually in the four quadrants at 1 and 1.5 mm from the OD. We considered the inner retina (IR) as the space between internal and external limiting membranes (ILM, ELM), the outer retina (OR) between the ELM and Bruch’s membrane (BM), and the outer nuclear layer (ONL) between the outer plexiform layer (OPL) and the ELM. Blue autofluorescence fundus images were acquired to inspect possible metabolic stress in the retinal pigmented epithelium (RPE). The Spectralis OCTA Module was used to capture 3D pictures of the retinal superficial and deep vascular plexuses. When the same eye needed to be examined repeatedly, e.g., during aging, we used the same equipment with the same angle of view as in the first scan. Then, we selected the first scan as the reference image and started the ‘Follow-up’ function of the Spectralis software. With the help of typical fundus landmarks such as the optic disc and vasculature pattern, we positioned the animal head in the same orientation and set the same focus as in the reference image.

### 2.3. ERG

Full-field ERG was recorded in the mice placed inside a ganzfeld globe (Figure 1B) controlled by RETIanimal system (Roland Consult), with the head fixed in a simple stereotaxic holder. The active golden ring electrodes were gently, i.e., without pressing against the eye, positioned on the cornea so that the center of the electrode was aligned with the center of the pupil. The reference and grounding needle electrodes were inserted subdermally at the center line near the tip of the snout and towards the tail base, respectively. Impedances were typically around 7.5 kΩ for the active electrodes and around 5 kΩ for the reference electrode, respectively. Signals were band-pass filtered between 1 and 300 Hz and recorded with 1024 samples/s resolution. Stimulation was repeated 20 times for each light intensity and individual responses were then averaged. For the scotopic protocol, mice were dark-adapted overnight in red individually ventilated cages (Tecniplast) in a dark room and all manipulations preceding the actual recording were done under dim red light. Stimuli were delivered as single flashes of luminance between 0.001 and 10 cd.s.m^−2^ distributed in 9 exponential steps. Before the photopic protocol, animals were exposed to 30 cd.m^−2^ white light for 2 min and recording continued with this background illumination. Stimulation consisted of single flashes between 1 and 100 cd.s.m^−2^ in 5 intensity steps. Signals were exported as csv files and analyzed using custom scripts in Matlab (MathWorks, Natick, MA, USA). Amplitudes and implicit times (i.e., latencies) of waves a and b were quantified; for details, see Figure 3.

### 2.4. Statistical Analysis

Statistical analysis and data plotting were done in GraphPad Prism (GraphPad Software, San Diego, CA, USA). Left and right eyes were both included in the analyses for each mouse, i.e., without averaging between sides. The *p*-value limit considered as significant was 0.05.

## 3. Results

### 3.1. Eye-Screening Procedure

Between the years 2018 and 2022, we imaged approximately 7500 mice with OCT, out of which 7000 were examined within the IMPC systemic phenotyping project and represented around 300 different knockout lines (an overview may be found in the Appendix A). During the same period, we recorded the ERG in more than 600 mice, of which 250 were suggested for ERG testing based on their primary OCT results and represented around 30 IMPC knockout lines. They were subjected to ERG examination within the same week that the OCT was performed, i.e., 15 weeks old. Several mouse lines displayed interesting eye phenotypes during primary screening; here, we present two examples with the Xrcc5 and Crx genes knocked out, respectively, that were examined in more detail. The complete primary phenotyping results, including findings from other tests than the examination of vision, can be found in the IMPC database (https://www.mousephenotype.org (accessed on 1 December 2022)).

### 3.2. Xrcc5^−/−^

The homozygous knockout mice were dramatically smaller than the controls, their weight was approximately half compared to the age-matched wild-types, and a substantial proportion of the animals did not survive until the age of the vision screen. Retinal assessment by OCT was carried out in five homozygous Xrcc5 knockouts and five control wild-type mice (10 + 10 eyes) at the age of 15 weeks. An example of an Xrcc5^−/−^ fundus is shown in Figure 4. Blue autofluorescence imaging reveals spotty lesions, and retinal cross-sections show peripheral retinal detachment. Abnormalities in the superficial vascular plexus morphology and white spots in the autofluorescence images appeared in all eyes, whereas sporadic cases of the retinal detachment were found only in 3 out of 10 eyes. The total retinal thickness in Xrcc5^−/−^ was significantly reduced in all parts of the fundus when compared to controls (Figure 5). The most affected region was the nasal quadrant; nevertheless, in comparison to the other quadrants, this was not statistically significant. The cross-sections were manually measured and thinning was observed in the inner retina, specifically in the ONL (49.8 ± 2.9 µm, 56.6 ± 1.3 µm in control animals) and in the ganglion cell/internal plexiform layers (55.3 ± 1.7 µm in the mutants compared to 65.6 ± 2.0 µm in controls). We recorded ERG in the same five homozygous knockout mice (10 eyes) three days after OCT and compared it to our internal database of control wild-type data (96 eyes at the time of analysis), as seen in Figure 6. Knockouts showed a significant decrease in the amplitude of the ERG waves, except for the photopic a-wave, i.e., the response of the cones. Latencies of the ERG waves were prolonged, which was consistent for all waves, including the cone response. Selected OCT and ERG parameters are summarized in Table 1.

### 3.3. Crx^−/−^

This strain was characterized by an extreme retinal degeneration accompanied by an almost complete loss of photoreceptors. At the age of 15 weeks, large patches in the fundus were spread over the whole retina and the images of blue autofluorescence revealed numerous small hyperreflective spots in all 16 examined mice of both sexes (example in Figure 4A, right). The total retinal thickness was reduced to 124.8 ± 9.1 µm. The innermost layers (nerve fibers, ganglion cells and IPL) were fused together and only the INL remained well visible in the cross-sectional images (Figure 4B, right). The distance between the ILM and OPL decreased to less than 80 µm. The ONL remnants were hardly detected and its maximum thickness was around 10 µm. The outer photoreceptor segments were missing. The ERG confirmed this severe pathology; there was no response to light stimulation even for the highest intensity of light (Figure 4E). One of the Crx^−/−^ mice was re-examined by OCT at the ages of 33, 50, 60, and 76 weeks. The gradual degeneration of all layers continued and the total retinal thickness eventually decreased to 93.7 ± 11.5 µm at 76 weeks of age. Further disorganization and dramatic reduction affected also the INL, which, in the end, was only partially visible in a few cross-sections, and its thickness dropped to less than 15 µm.

### 3.4. OCT and ERG during Aging

Wild-type C57Bl/6NCrl mice of both sexes were examined by OCT and ERG at the ages of 10, 15, 30, 50, 70, and 100 weeks. An example cross-sectional image and thickness map of the retina, along with the scotopic ERG response of the same mouse, in three different ages is shown in Figure 7. Retinal thickness in each quadrant was automatically analyzed by the Heidelberg software, as shown in Figure 8 and Table 2. The thickness of the retina was consistent between 10 and 70 weeks, and then a significant reduction was observed in all retinal quadrants in 100-week-old animals, as shown in Figure 8A. The most severe retinal thinning occurred in the inferior and nasal part of the retina (Figure 8B). The scotopic and photopic ERG amplitudes progressively declined with age (Figure 9). Statistical analysis (two-way ANOVA with Tukey‘s multiple comparisons, *p* < 0.05) proved a significant decrease between each two consecutive age groups except for the groups of 50 and 70 weeks, which showed almost identical data values. This finding was consistent at least for the highest intensity of stimulation for both, the scotopic and photopic responses, respectively. The analysis of the implicit times was less consistent and returned a significant outcome only sporadically, with no obvious pattern related to the age groups.

## 4. Discussion

### 4.1. Eye-Screening Procedure

We have demonstrated a working scheme optimized for a high-throughput phenotyping where the OCT is the primary approach and the ERG test follows only if the primary findings justify it. The OCT protocol itself is relatively fast; the maximal daily capacity in the case that no further examinations need to be done is around 22 mice, with both eyes tested, including an examination of the front eye segment, which is not discussed in this text. The basic OCT screen may be promptly complemented with additional techniques while the mouse is still anesthetized if the fundus image shows abnormalities. When the blue autofluorescence and angiography were required for each animal, the maximal capacity decreased to 14–15 mice per day. Nevertheless, the most time-consuming step was the analysis of the retinal thickness. The recording software offered a fully automated algorithm for the detection of individual layers in the retina and an estimation of its thickness. However, the algorithm had most probably been optimized for a human eye and the results deviated from values expected for a mouse retina. We compared three approaches to the estimation of the retinal thickness, see Figure 5 and Figure 2B. First, while accepting the automatic measurement was the fastest solution, it resulted in many nonsense detections of the outer retinal boundary away from the retina. Second, we corrected these obvious errors in order to eliminate the most outlying alignments. The third, most time-consuming, method was a fully human-controlled analysis where the experimenter manually positioned the boundaries of the retina in selected cross-sections and measured values in four quadrants and two distances from the optic disc. A comparison in the Table 1 shows that, despite the difference in the absolute values, a robust phenotype, represented here by a Xrcc5 knockout line, always returned a statistically significant finding regardless of the method used.

Electrophysiology was measured under two distinct conditions. First, mice were adapted to the dark overnight in order to obtain scotopic responses. The scotopic ERG reflects rod-driven responses at the lowest stimulation intensities and a mixed rod–cone vision when moderate and strong stimuli are used (reviewed in [7]). The dark adaptation period needs to be taken into consideration when planning the time schedule of the ERG experiments. Next, in order to distinguish pure cone-driven retinal function, rods were desensitized by exposing the animals to moderate light for two minutes after the scotopic examination finished. The ERG recording protocol itself was limited by the time window of the general anesthesia duration to approximately 45 min, including anesthesia induction, the time necessary for a complete pupil dilation (15 min together), and the placing of the electrodes (5–10 min). However, when multiple mice were tested in a row, an application of anesthetics and atropine to the next mouse could start 15 min before the end of the previous recording. Thus, the time interval between individual experiments shortened to 30 min. Considering only one device, an 8 h working day, 30 min of preparation at the start, and at least 80 min needed for the last mouse to fully wake from the narcosis, we estimate the maximal capacity of the described protocol to be 12 mice/day, without data analysis. In practice, however, the continuity of the progress may be interrupted by technical issues or uncollaborative mice; in our experience, 9–10 mice/day is a realistic average efficiency. The ERG recording protocol comprised a scotopic single-flash stage (10^−3^–10^1^ cd.s.m^−2^), a photopic single-flash stage (1–100 cd.s.m^−2^), 400 ms long pulse photopic stimulation, and a photopic flicker stimulation (2–40 Hz). The results collected from the flicker and long-pulse protocols, however, were not provided here, since not all the mice presented here went through the entire protocol. Nevertheless, we show that the ERG can be efficiently used as a secondary testing method alongside the OCT, and the battery of tests may cover the majority of the ERG procedures suggested in the ISCEV recommendations for the human clinical ERG [20,21,22,23,24].

### 4.2. Xrcc5^−/−^

The Xrcc5 gene encodes the Ku80 subunit of the Ku heterodimer protein, also known as DNA repair protein Xrcc5. Mutations in this gene are associated with serious diseases such as Werner Syndrome (WS), which is characterized by premature aging and malignant cancer predispositions [25,26,27]. One WS symptom is an eye cataract at a young age. Among the ten eyes of the five knockout mice, we observed only one case of a cataract; on the other hand, all had increased opacity of the cornea. We found a decrease of the Xrcc5^−/−^ central retina thickness as a consequence of a thinner ONL, which also occurs in multiple forms of retinitis pigmentosa. ONL is also a location with a high expression of WRN, another gene associated with WS [28]. Compared to the controls, we observed prolonged implicit times and smaller amplitudes of the ERG waves in the dark-adapted Xrcc5^−/−^ eyes, which signifies a disturbed function of rods and rod-related retinal processing. The light-adapted ERG showed similar changes with one exception, the amplitude of cone response, i.e., the a-wave, remained unchanged. This suggests that cones might be less affected by the mutation than the rods. Interestingly, it was recently reported that Ku80 is expressed in the terminals of mouse rods but not in the cones [29]. We also noticed several cases of retinal detachment in the periphery of the knockout eyes, which can be an indication of preliminary aging as another symptom of WS.

### 4.3. Crx^−/−^

The Crx gene encodes photoreceptor-specific transcription factor important for cone and rod differentiation and maintenance [30]. Mutations in this gene result in progressive photoreceptor degeneration with progressive loss of vision, such as in retinitis pigmentosa, Leber congenital amaurosis type III, and the autosomal dominant cone–rod dystrophy 2 [31,32]. The test of vision in 15-week-old Crx^−/−^ mice within the IMPC screen confirmed a dramatic reduction (about 50%) in the retinal thickness compared to the controls due to the thinning of the ONL and the missing outer segments of the photoreceptors. Consequently, there was no response to light stimulation detectable in the ERG. Furukawa [30] reported a progressive loss of cell rows in the ONL until, at the age of 6 months, only one to three remained. We followed one representative animal until the age of 1.5 years and observed that the ONL had completely disappeared and the INL was visible only as sporadic remnants, with a maximal thickness of 15 µm.

### 4.4. OCT and ERG during Aging

We observed a gradual decrease of scotopic and photopic ERG amplitudes in wild-type C57Bl/6NCrl mice between the age of 10 and 100 weeks. The only exception was a temporary plateau between 50 and 70 weeks. Responses of both, photoreceptors (a-wave) and ON bipolar cells (b-wave), were equally reduced, i.e., approximately to 60% in 50-week-old mice, and they dropped further below 30% at the age of 100 weeks in comparison to the data obtained at 10 weeks. The bipolar cells are first neurons with direct synaptic contacts to photoreceptors; hence, the b-wave reduction can be interpreted as a simple consequence of reduced input from the photoreceptors. The decline in the mouse ERG during aging was previously interpreted as a consequence of the reduced number of photoreceptors [33], tissue-specific metabolic changes [34,35], or the shortening of the photoreceptor outer segments and lower content of opsin [36]. Potentially, these factors may also affect the dimensions of individual retinal layers; however, our analysis of retinal thickness derived from the OCT by automated segmentation did not show any changes until the age of 70 weeks. Surprisingly, it significantly dropped in the oldest, 100-week-old, animals. These results are consistent with previously published OCT data obtained by a similar method (i.e., semiautomated segmentation by Heidelberg Spectralis software) in C57Bl/6J mice [37], where retinal thickness in wild-type mice did not change between 6 and 20 months of age. Small but progressive retinal thinning with increasing age was, however, reported by authors who quantified the thickness entirely manually [38]. Moreover, it was shown in [39] that different techniques of retinal segmentation, automated or manual, led to different absolute values of the retinal thickness. We failed to correlate the automatically measured retinal thickness with the ERG amplitudes (plot not shown), but we cannot exclude that a manual analysis of the retina or, specifically, ONL, would show a correlation with the ERG, as was shown in [36].

Our study was performed on C57Bl/6NCrl mice, an inbred strain that serves as a control and the background strain for production of mutant mouse lines in the majority of IMPC centers. All the C57Bl/6N substrains are known to carry the rd8 mutation in the Crb1 gene [40]. A mild focal degeneration in the retina of adult rd8 homozygous mice has been previously described [11,41,42] and was accompanied by changes in suprathreshold ERG parameters at the age of 9 months and higher [42,43]. Due to this, it is probable that the decreasing size of ERG responses we observed with aging has not resulted exclusively from age-related changes in the retina alone; rather, the presence of the rd8 mutation in the ‘NCrl’ mouse substrain might have also played a role. Nevertheless, studies that used mouse strains unaffected by the rd8 mutation also showed a serious decrease in the ERG amplitudes in mice that were around 1-year-old or older [33,34,35,36,44].

## 5. Conclusions

We have demonstrated that, in the high-throughput screening of mouse vision, primary OCT examination can be complemented with ERG in justified cases without compromising the time schedule of the planned experiments. Retinal thickness can be retrieved from OCT images using an automated segmentation algorithm while taking into consideration that the quantification is affected by the imprecise detection of the retinal outer boundary. An automated approach can be used to prove a robust pathological phenotype in the retina, and we describe two examples of such phenotypes in Crx and Xrcc5 knockouts, respectively. Nevertheless, the detection of subtle changes in the retinal thickness, such as a potential slight thinning at a higher age, may require manual segmentation. We report a gradual decline in the ERG amplitudes between the ages of 10 and 100 weeks in the C57Bl/6NCrl mouse strain, which was not accompanied by changes in the total retinal thickness until a sudden drop between 70 and 100 weeks of age.

## Figures and Tables

**Figure 1 genes-14-00294-f001:**
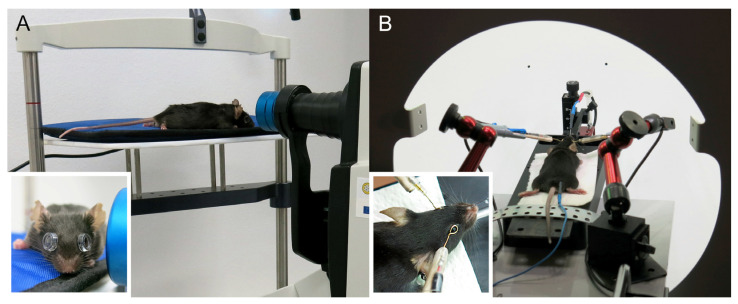
OCT and ERG setup. (**A**) A mouse with contact lenses lying on a modified human headrest in front of SD-OCT camera which was adapted to animal experiments by an additional +25 dioptric objective lens. Inlay: OCT in mice requires use of +100 dioptric contact lenses; (**B**) During ERG recording, the mouse was inside a ganzfeld globe with the head fixed by the upper incisors to a simple stereotaxic holder and positioned so that the eyes were approximately in the center of the globe. Inlay: close-up view of the golden wire active electrodes placed on eyes.

**Figure 2 genes-14-00294-f002:**
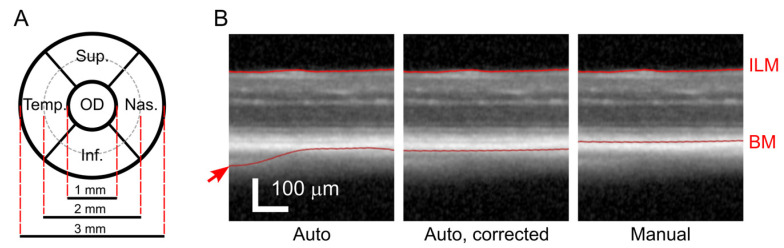
Measurement of the retinal thickness from OCT. (**A**) A scheme of divisions of the scanned retinal area exemplified on the right eye. The inner-most circle, optic disc (OD), contains the head of the optic nerve and was excluded from quantifications. The surrounding area was subdivided into four quadrants: superior (Sup.), inferior (Inf.), nasal (Nas.), and temporal (Temp.), respectively. The quadrants were further split into outer and inner parts (inner and outer ring, dashed line) which were quantified separately. In this text, we report the retinal thickness in each quadrant as an average of the outer and inner segment; (**B**) Three approaches to the retinal thickness measurement. The OCT recording software offered a fully automated detection of the retinal boundaries, i.e., the inner limiting membrane (ILM) and the Bruch’s membrane (BM), respectively (Auto, left image). Since substantial errors appeared in the automatic results (arrow), we corrected the obviously outlying points (Auto, corrected, middle image). The most precise and time-consuming approach, however, required a manual setting of the BM to its correct position above choriocapillaris (Manual, right image).

**Figure 3 genes-14-00294-f003:**
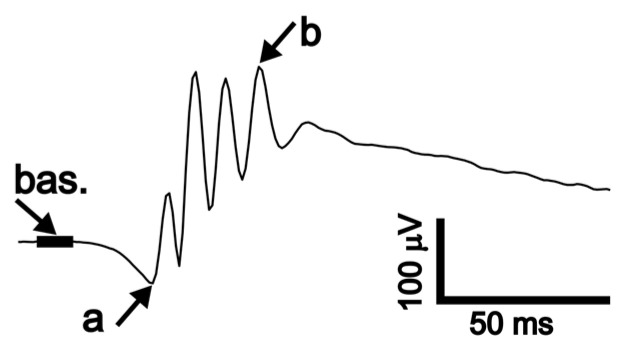
Measurement of ERG waves. ERG was analyzed by a custom script (Matlab) with additional visual inspection of the correctness of the wave detection. Baseline value (bas.) was calculated as an average of 10 ms of the signal preceding stimulation. The amplitude of wave a (a) was measured between baseline and a local minimum in the signal. The amplitude of wave b (b) was measured between the a-wave trough and the local maximum of the signal.

**Figure 4 genes-14-00294-f004:**
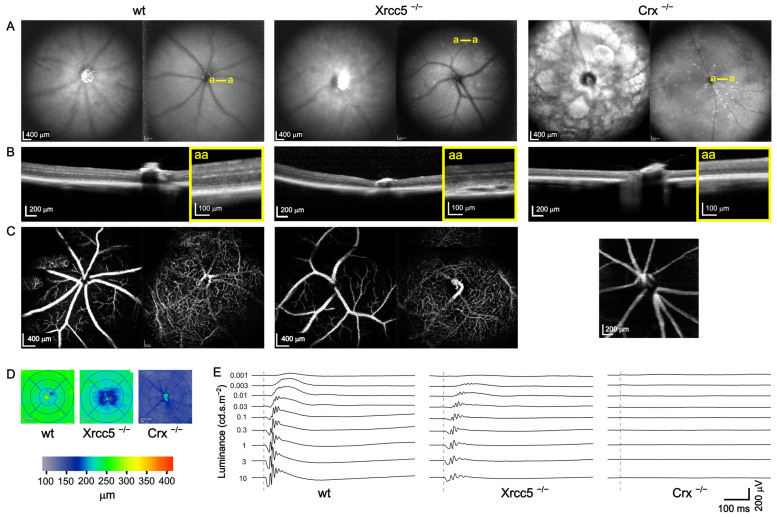
Example of OCT and ERG in a wt control mouse and Xrcc5 and Crx homozygote knockouts. (**A**) Representative SD-OCT infrared fundus images (left) and blue autofluorescence (BAF, right) images for each mouse strain. Large spotty lesions and an atypical pattern of superficial blood vessels were visible in Xrcc5^−/−^. Severe retinal degeneration in the Crx^−/−^ mouse was accompanied by a typical pattern of large white patches in the fundus and many small white and dark dots in the BAF; (**B**) Cross-sectional images of the retina through the head of the optic nerve with zoomed-in views (yellow framed squares) of selected (yellow a–a lines in BAF) retinal parts. Peripheral retinal detachments were observed in Xrcc5^−/−^ mice, see the magnified view; (**C**) Superficial (svp) and deep (dvp) vascular plexuses. Svp (left image) of the Xrcc5^−/−^ mouse displayed an atypical pattern with crossings and branching of blood vessels, whereas the dvp (right image) was similar to the wt. On the other hand, in the Crx^−/−^ animals, the retina was too thin to distinguish the border between the svp and dvp. Instead, we present one merged image of the whole retina depth that appears to contain no dvp; (**D**) Automatic retinal thickness quantification. A severe decrease in the retinal thickness was observed in the Crx^−/−^ animal; (**E**) Typical scotopic ERG responses of the three mouse lines plotted to scale. Dashed lines denote the moment of light flash. The Xrcc5^−/−^ mouse had smaller responses than the wt. The Crx^−/−^ animal completely lacked any ERG response.

**Figure 5 genes-14-00294-f005:**
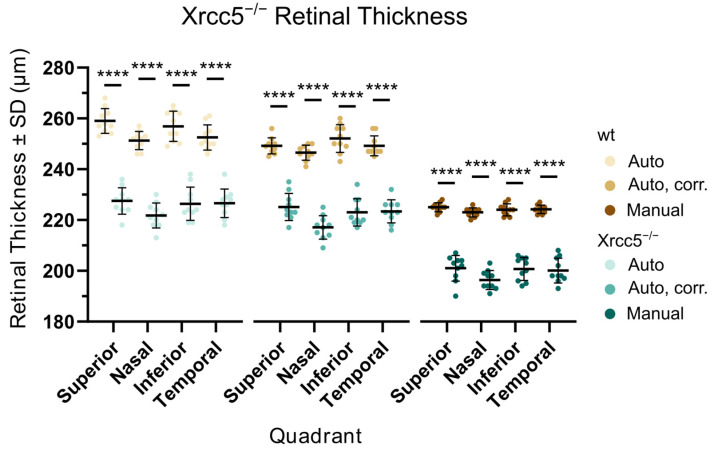
Retinal thickness in individual retinal quadrants for Xrcc5^−/−^ and wt controls. The thickness of the retina was quantified using three different approaches. The first (left, Auto) was an automated measurement performed by the recording software with no additional postprocessing. In the second (middle, Auto, corr.), we corrected the automatically created retinal segmentation to eliminate the most misaligned values. The third (right, Manual) represents measurements taken from manually set retinal boundaries. For details, see Figure 2B. All three approaches confirmed a highly significant (2-way ANOVA, *p* < 0.0001, marked by ****) decrease of retinal thickness in the mutants.

**Figure 6 genes-14-00294-f006:**
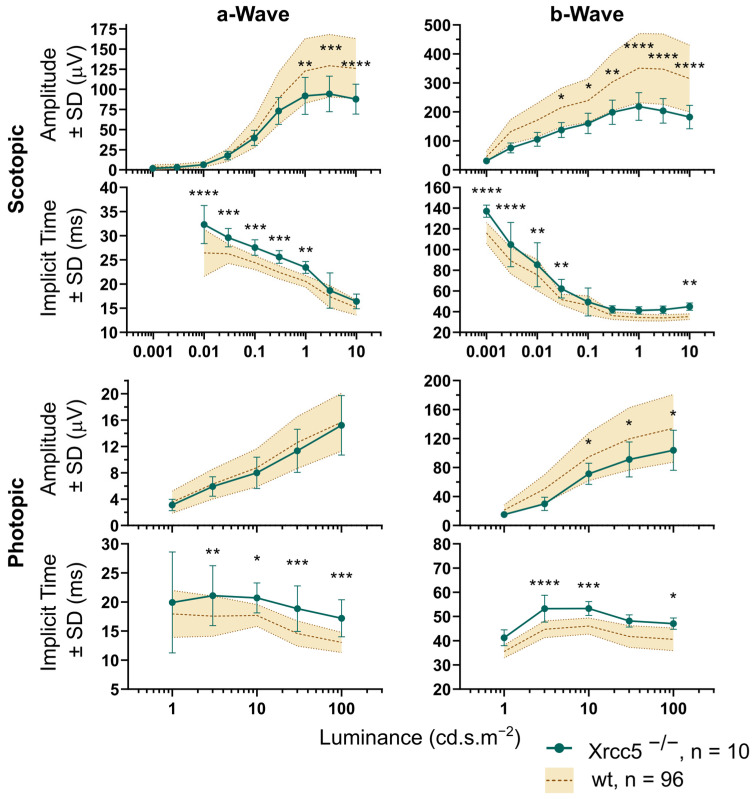
Scotopic and photopic ERG of Xrcc5^−/−^ mice. Amplitudes and implicit times of dark-adapted waves a and b and light-adapted waves a and b of the Xrcc5 homozygote knockouts compared to the set of control wild-type (wt) data. Amplitudes and implicit times share the same x-axis, the luminance of stimuli. In general, the amplitudes of the Xrcc5^−/−^ were significantly smaller compared to the controls (2-way ANOVA with Šídák multiple comparisons) for the highest stimulation intensities, except for the photopic wave a. Decrease of amplitude was also accompanied with prolonged implicit times of the respective waves. The *p* value is marked by asterisks, * *p* < 0.05, ** *p* < 0.01, *** *p* < 0.001, **** *p* < 0.0001, n represents the total number of eyes included in the analysis.

**Figure 7 genes-14-00294-f007:**
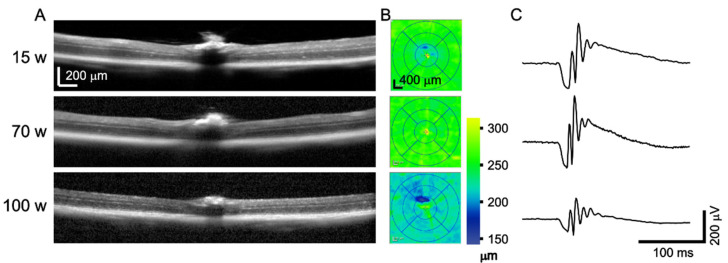
Example of OCT and ERG of the same mouse in the age of 15, 70, and 100 weeks. (**A**) Representative cross-sectional images through the optic disc; (**B**) Retinal thickness quantification based on automated retinal segmentation with minor manual corrections; (**C**) Dark-adapted ERG response to a single-flash stimulation of 3 cd.s.m^−2^ luminance.

**Figure 8 genes-14-00294-f008:**
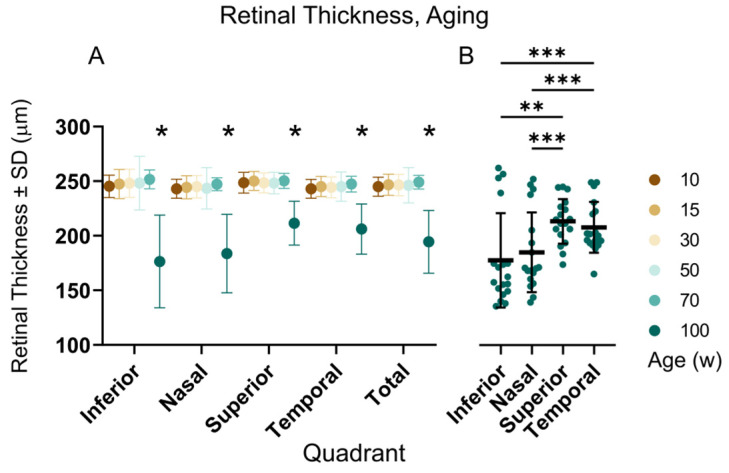
Retinal thickness during aging quantified by automatic retinal segmentation. (**A**) Retinal thickness did not change between 10- and 70-week-old mice in any quadrant of the retina. The only significant decline (2-way ANOVA with Tukey’s multiple comparisons, *p* < 0.0001) was observed in the oldest, 100-week-old individuals; (**B**) Further analysis of the elderly animals revealed that the most affected areas of the retina were the inferior and nasal quadrants (2-way ANOVA with Tukey’s multiple comparisons, * *p* < 0.05, ** *p* < 0.01, *** *p* < 0.001. The y-axis is shared with (**A**).

**Figure 9 genes-14-00294-f009:**
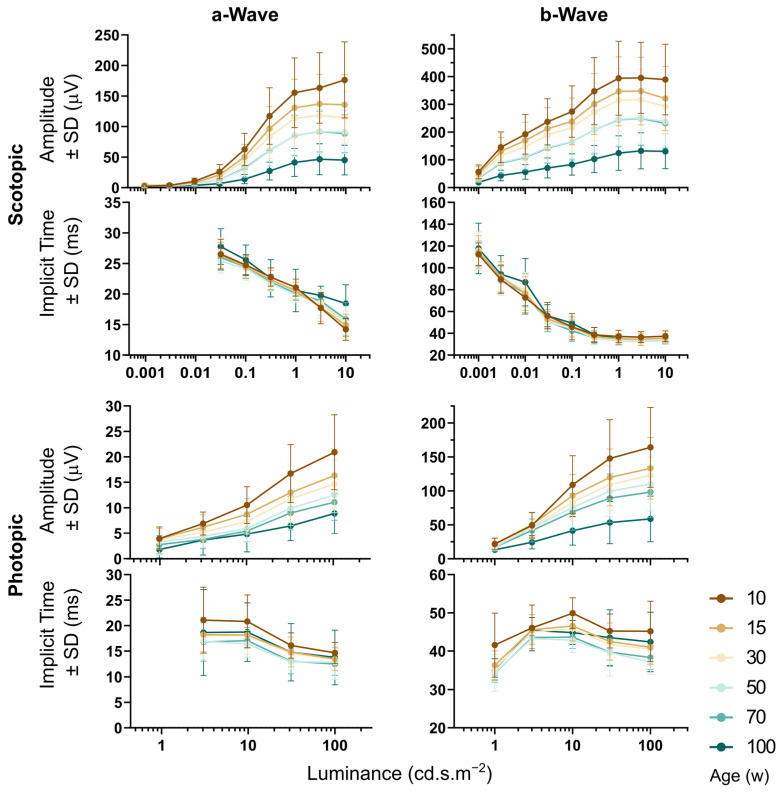
Full-field single-flash ERG during aging. Scotopic (upper part) and photopic (lower part) a and b waves, respectively. Related plots share the same x-axes (luminance of the stimuli). Decrease in response amplitude was observed between each two consecutive age groups except for the groups of 50 and 70 weeks. This finding was consistent for both scotopic and photopic responses. Age-related changes in implicit times were not obvious and the analysis returned a significant outcome only sporadically. The implicit times of the responses to the lowest luminance values were omitted from the analysis because the batch algorithm used was unable to correctly detect it when response amplitude was close to the background noise.

**Table 1 genes-14-00294-t001:** Selected OCT and ERG parameters for Xrcc5^−/−^ and control (wt) mice. Values are presented as (mean ± SD, number of observations).

Parameter	Unit	Wt (Mean ± SD, n)	KO (Mean ± SD, n)	*p*
Total retinal thickness, autosegmentation	µm	254.9 ± 3.1, 10	225.6 ± 4.2, 10	<0.0001
Total retinal thickness, auto with corrections	µm	249.3 ± 3.0, 10	222.2 ± 3.7, 10	<0.0001
Total retinal thickness, manual measurement	µm	224.1 ± 1.3, 10	199.6 ± 2.5, 10	<0.0001
Scotopic a amplitude, 10 cd.s/m^2^	µV	126.0 ± 37.1, 96	87.7 ± 18.7, 10	<0.0001
Scotopic b amplitude, 10 cd.s/m^2^	µV	314.9 ± 115.5, 96	181.9 ± 40.5, 10	<0.0001
Photopic a amplitude, 100 cd.s/m^2^	µV	15.7 ± 4.4, 96	15.2 ± 4.5, 10	0.9956
Photopic b amplitude, 100 cd.s/m^2^	µV	134.1 ± 46.7, 96	103.8 ± 27.7, 10	0.0245
Scotopic a implicit time, 10 cd.s/m^2^	ms	15.1 ± 1.5, 96	16.4 ± 1.5, 10	0.5465
Scotopic b implicit time, 10 cd.s/m^2^	ms	35.1 ± 2.8, 96	44.8 ± 3.6, 10	0.0086
Photopic a implicit time, 100 cd.s/m^2^	ms	13.0 ± 1.7, 96	17.2 ± 3.2, 10	0.0003
Photopic b implicit time, 100 cd.s/m^2^	ms	40.6 ± 4.7, 96	47.1 ± 2.4, 10	0.0248

**Table 2 genes-14-00294-t002:** Retinal thickness and selected ERG parameters for 6 different ages. Values are presented as (mean ± SD, number of observations). Stimulation luminance is indicated, and it was 10 and 100 cd.s/m^2^ for the scotopic and photopic condition, respectively.

Parameter	Unit	10 w	15 w	30 w	50 w	70 w	100 w
Total retinal thickness, auto-segmentation	µm	244.9 ± 9.1, 74	246.7 ± 10.3, 66	246.4 ± 10.2, 81	246.2 ± 16.5, 22	249.0 ± 7.0, 36	194.4 ± 30.0, 18
Scotopic a amplitude, 10 cd.s/m^2^	µV	176.3 ± 62.3, 143	135.7 ± 49.2, 403	114.9 ± 42.1, 153	91.0 ± 35.3, 50	88.6 ± 30.9, 78	45.1 ± 24.2, 31
Scotopic b amplitude, 10 cd.s/m^2^	µV	389.0 ± 127.2, 143	321.0 ± 115.3, 403	292.2 ± 99.4, 153	236.9 ± 86.5, 50	231.6 ± 89, 78	130.3 ± 62.5, 31
Photopic a amplitude, 100 cd.s/m^2^	µV	20.8 ± 7.3, 88	16.3 ± 5.1, 358	14.6 ± 4.6, 136	12.4 ± 3.4, 46	11.0 ± 3.5, 64	8.8 ± 3.9, 20
Photopic b amplitude, 100 cd.s/m^2^	µV	164.2 ± 58.9, 88	133.2 ± 45.2, 358	123.0 ± 40.2, 136	110.1 ± 39.8, 46	98.3 ± 37.7, 64	58.7 ± 33.7, 20
Scotopic a implicit time, 10 cd.s/m^2^	ms	14.2 ± 1.7, 143	14.8 ± 1.8, 403	15.2 ± 1.9, 153	15.3 ± 2.5, 50	15.8 ± 2.7, 78	18.4 ± 3.0, 31
Scotopic b implicit time, 10 cd.s/m^2^	ms	37.3 ± 4.9, 143	35.5 ± 4.5, 403	34.7 ± 2.5, 153	33.1 ± 2.8, 50	33.5 ± 3.4, 78	34.9 ± 3.0, 31
Photopic a implicit time, 100 cd.s/m^2^	ms	14.6 ± 2.0, 88	13.4 ± 2.2, 358	13.4 ± 2.0, 136	12.8 ± 3.3, 46	12.4 ± 2.1, 64	13.7 ± 5.3, 20
Photopic b implicit time, 100 cd.s/m^2^	ms	45.2 ± 7.9, 88	41.0 ± 4.4, 358	40.6 ± 3.8, 136	37.1 ± 3.2, 46	38.3 ± 3.2, 64	42.4 ± 7.8, 20

## Data Availability

Primary phenotyping data for Crx and Xrcc5 knockout mouse lines is openly available from the IMPC webpage: www.mousephenotype.org (accessed on 1 December 2022).

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
