# Peer review of "OCT and ERG Techniques in High-Throughput Phenotyping of Mouse Vision"

_genes, 2023, doi:10.3390/genes14020294_

Round 1
Reviewer 1 Report
This paper, "OCT and ERG techniques in high-throughput phenotyping of mouse vision," by Lindovski et. al. originates with the rationale that phenotyping tests should be organized into laboratory animal model characterization and screening. The authors have used commercially available clinical grade ERG and OCT machines that has been modified to test laboratory mouse. Clinical characterization thus parallels human studies and findings are thus relevant. The paper is logically presented and to educate vision science world about the use of these techniques as standard measures. I have several suggestions to improve the manuscript presentation.
1) The authors say that they have evaluated a total of 7000 rodents till date in preparation for this paper yet the findings are presented for only a few animals in each category 10 + 10 eyes for Xrcc5 mice.
2) The description of ERG measurement protocol is not adequate as the average amplitude of b-wve is very low. Typically for a wildtype rodent or a healthy individual the b-wave amplitude is more than twice that of the a-wave amplitude. In the present study, even for the wildtype mice b-wave is mildly more than the measured a-wave amplitude. As I understand from the Figure 3, the authors have not included OP into the amplitude of the b-wave. The authors should revise the data as its standard for the vision community.
3) Although the b-wave amplitude is understandable, the c-wave amplitude is not acceptable given that the authors are using a ganzfeld stimulator that can elicit c-waves in amplitudes larger than the b-wave. Even when used 10 cd.s/m^2 in table 2
4) Again this reviewer failed to follow the logic of comparing 1 cd.s/m^2 in table 1 but 10 cd.s/m^2 in table 2.
5) In the aging group if the ERG response is related to number of photoreceptors, the authors should adequately discuss, the severe reduction in b- and c-wave response. How did this correlate with OCT findings.
6) I would encourage increasing the size of OCT images to appreciate any difference in structure between the groups.
Reviewer 2 Report
In this manuscript, Lindovský et al. demonstrated their recent work on high-throughput screening of mouse vision with OCT, autofluorescence, and ERG measurements. Quantitative analysis of retina thickness and ERG wave changes were shown for different mouse models, which are very meaningful for understanding connections between specific retina structure and/or functions with the gene expression.
I have a few minor comments listed below.
Line 41-43: The Heidelberg Engineering system is actually a multimodality imaging system, with OCT, OCTA and blue autofluorescence (and more other modes not mentioned in the manuscript). However, autofluorescence detection imaging is not obtained through the OCT channel, instead, it is the confocal detection channel of the Heidelberg Engineering system. Please rephrase the statement here to differentiate the OCT imaging channel and autofluorescence imaging channel.
Line 59-63: Please add relevant references of the mouse eye structure and composition.
Line 146: Please add description of physical or physiological meanings of a, b, and c waves in the relevant text.
Line 212: In Figure 6, given a specific luminance, multiple ERG measurements were done. Do differences (see Figure 1B) among measurements such as locations where active and reference electrodes were placed, and the force/pressure applied when electrodes were attached to the cornea, influence the final results?
Line 229: Figure 4A should be cited here instead of Figure 3A.
Line 231: Figure 4B should be cited here instead of Figure 3B.
Line 235: Figure 4E should be cited here instead of Figure 3E. Also please check other figure citations.
Line 259: Figure 7. Complete co-registration of OCT images at different time points can be difficult. The field of view, depth location of the imaging focus, orientation of the retina relative to the illumination beam, etc. might not be exactly the same. Please elaborate more on how to carefully avoid this inconsistency to make the analysis more quantitative.
Line 283: Table 2 does not show up properly.
Discussion panel: There are thorough measurements and analysis of ERG behaviors of both the scotopic and photopic protocols. Please elaborate more on the contrasts between the two cases.
Round 2
Reviewer 1 Report
Thank you for responding to my review and the paper is stronger. I find two noticeable difference between the initial submission and the present version. For both new Figures 6 and 9 the b-wave implicit time trend is much different than the previous figure. Also, please explain the photopic results in both tables. It appears that there is no reduction in a-wave but slight reduction in b-wave amplitude.
